# Synthesis and Antibiotic Activity of Chitosan-Based Comb-like Co-Polypeptides

**DOI:** 10.3390/md21040243

**Published:** 2023-04-15

**Authors:** Timothy P. Enright, Dominic L. Garcia, Gia Storti, Jason E. Heindl, Alexander Sidorenko

**Affiliations:** 1Department of Chemistry and Biochemistry, Saint Joseph’s University, Philadelphia, PA 19104, USA; te20345584@sju.edu (T.P.E.);; 2Department of Biological & Biomedical Sciences, Rowan University, Glassboro, NJ 08028, USA; heindl@rowan.edu

**Keywords:** antimicrobial peptides, chitosan, comb-like co-polypeptide, *N*-carboxyanhydrides, ring-opening polymerization

## Abstract

Infections caused by multidrug-resistant Gram-negative bacteria have been named one of the most urgent global health threats due to antimicrobial resistance. Considerable efforts have been made to develop new antibiotic drugs and investigate the mechanism of resistance. Recently, Anti-Microbial Peptides (AMPs) have served as a paradigm in the design of novel drugs that are active against multidrug-resistant organisms. AMPs are rapid-acting, potent, possess an unusually broad spectrum of activity, and have shown efficacy as topical agents. Unlike traditional therapeutics that interfere with essential bacterial enzymes, AMPs interact with microbial membranes through electrostatic interactions and physically damage cell integrity. However, naturally occurring AMPs have limited selectivity and modest efficacy. Therefore, recent efforts have focused on the development of synthetic AMP analogs with optimal pharmacodynamics and an ideal selectivity profile. Hence, this work explores the development of novel antimicrobial agents which mimic the structure of graft copolymers and mirror the mode of action of AMPs. A family of polymers comprised of chitosan backbone and AMP side chains were synthesized via the ring-opening polymerization of the *N*-carboxyanhydride of l-lysine and l-leucine. The polymerization was initiated from the functional groups of chitosan. The derivatives with random- and block-copolymer side chains were explored as drug targets. These graft copolymer systems exhibited activity against clinically significant pathogens and disrupted biofilm formation. Our studies highlight the potential of chitosan-*graft*-polypeptide structures in biomedical applications.

## 1. Introduction

Antimicrobial peptides (AMPs) are a unique class of peptides that are part of the innate immune response of most multicellular organisms [1]. Unlike traditional antibiotics, which inhibit an intracellular target, many AMPs and their synthetic analogues rely on physically damaging the bacterial cell membrane [2]. As such, they offer promise as a versatile therapeutic to combat the rising trend of antimicrobial resistance. A number of AMPs with a broad range of structural and chemical compositions have been identified [3]. However, several characteristics are conserved [4]: effective AMPs possess a net positive charge, hydrophobic moieties, and amphiphilicity. Some approaches for improving the pharmacological performance of naturally derived AMPs have included chemical modification [5], non-native amino acids [6], fully or semi-synthetic scaffolds [7,8,9,10], supramolecular assemblies [11,12], and multimeric structures [13,14]. The rational design of synthetic analogues may facilitate further improvement of the pharmacodynamic properties. In particular, grafting peptides to a common polymer backbone to create an antimicrobial graft copolymer (GCP) may bring several advantages: (i) they provide an opportunity to incorporate various functionalities and tune parameters to achieve high activity, e.g., hydrophilic/hydrophobic balance; (ii) the graft copolymer structure provides a high local concentration of peptides which may facilitate bacterial membrane disruption thus reducing the minimally required global concentration compared to traditional AMPs; (iii) the redundancy of GCP’s peptide grafts may undergo conformational adjustment to maximize favorable interactions with the local environment and facilitate penetration of the GCP in the bacterial cell wall and promote cell death; (iv) the composition of the GCPs can be selected to complement the peptidoglycan layer of the cytoplasmic membrane of a bacteria. A complementary structure may improve selectivity over mammalian cells and promote adsorption onto the cell wall of bacteria [15]. Despite the large number of studies demonstrating the therapeutic potential of AMP-based therapeutics, only a few candidates have made it through the discovery phase and found clinical application [16]. 

In this work we report the synthesis of novel antimicrobial GCPs and evaluate their antibacterial activity. We used the natural polysaccharide chitosan (CHI) as a backbone to fabricate a small library of peptide graft copolymers via ring-opening polymerization (ROP) of *N*-carboxyanhydrides (NCA) of l-lysine and l-leucine. Graft copolymers with branched architectures composed of homo-, random, and block-polypeptide sequences were targeted (Figure 1). The graft copolymers were analyzed by NMR, FTIR, and SEC to characterize their molecular weight and composition. The results of an in vitro investigation of antibacterial activity against model pathogens and cytotoxicity for human dermal fibroblasts are reported. Finally, we assessed the effect of the GCPs on the biomass and cell viability within biofilms of *Agrobacterium tumefaciens*. 

## 2. Results and Discussion 

### 2.1. Synthesis of GCPs

The CHI backbone was prepared using a method adopted by Sashiwa et al. [17]. For complete dissolution to take place, the CHI-CSA salt was freeze-dried. The solid was found to readily dissolve in DMSO. With this development, further chemical modification of CHI could be carried out in homogeneous water-free conditions. Many factors in sourcing and manufacturing CHI affect the characteristics and composition of the final product. Since the amino groups of CHI initiate grafting, it is important to establish the fraction of available amino groups prior to modification. The ^1^H NMR spectrum of CHI-CSA in DMSO-d_6_ is presented in Figure 2. The signals at 3.5 ppm, 4.8 ppm, and 8.3 ppm are attributed to the −5, −1, and -NH_2_ hydrogens of CHI, respectively. The hydrogens bound to the pyranose ring (3, 4, 5, and 6) were assigned to a broad peak at about 3.8 ppm.

The degree of acetylation (*DA*) was calculated using a method proposed by Weinhold et al. where the area of the acetyl CH_3_ hydrogens $\left(A_{{CH}_{3}}\right)$, and the H2-H6 signals $\left(A_{H2\mathrm{-H}6}\right)$ of CHI are used according to Equation (1) [18]:(1)$$\mathrm{DA}=\left[1-\left(\frac{\frac{1}{3}\times A_{{\mathrm{CH}}_{3}}}{\frac{1}{6\left(A_{H2\mathrm{-H}6}\right)}}\right)\right]\times 100$$

The *DA* was found to be > 99%. Literature reports that *N*-acetyl linkages, as well as main chain (glycosidic) linkages of CHI, are susceptible to acidic hydrolysis [19]. It is likely that during the dissolution of CHI the acetyl groups are cleaved while in the presence of excess CSA. 

CHI-based GCPs could yield a versatile material with the combined characteristics of its components, tunable amphiphilicity, and potentially controlled interactions with the environment. Several research groups have synthesized such materials using the amino groups of CHI to initiate ROP of amino acid NCAs (Figure 3). However, the restricted solubility of CHI limits the number of synthetic pathways. Kurita et al. and Chi et al. synthesized CHI-*graft*-poly(γ-methyl l-glutamate) and CHI-*graft*-poly(l-lysine) using a biphasic interfacial approach in ethyl acetate and water [20,21]. Alternatively, the synthesis of CHI-*graft*-poly(l-glutamic) acid and poly(lysine-ran-phenylalanine) copolymers was accomplished in homogenous conditions using a soluble form of CHI, 6-*O*-triphenylmethyl CHI, in anhydrous DMF [22]. In this work, we adopted an approach first utilized by Perdith et al. [23] who synthesized CHI-*graft*-poly(sodium-l-glutamate) nanoparticles using the primary amine sulfonate salt, CHI-CSA, as a macroinitiator in DMSO. Extending this strategy, we explored the synthesis of CHI-based GCPs with hydrophobic l-leucine and cationic hydrophilic l-lysine amino acids.

The CHI-CSA was rapidly dissolved under argon in dry DMSO to which 0.3 molar equivalents of DIPEA with respect to protonated amines -NH_3_^+^ of CHI-CSA was added. The addition of DIPEA served to suppress acid-catalyzed cleavage of CHI and increase the propagation rate of NCA-ROP. Control over the reactivity of the growing polymer chain end is critical to achieve a high molecular weight with optimal conditions that are often necessary to be determined empirically. Amine salts such as CHI-CSA have diminished reactivity as a nucleophile due to the formation of the inactive protonated amines. Polymerizations initiated with hydrochloride salts were found to yield a single NCA addition and required elevated temperatures to proceed [24,25]. Since CHI degrades in the presence of sulfonic acid, we adjusted the equilibrium of free amines by increasing the alkali content instead of temperature (Figure 4). DIPEA was selected as it is a sterically hindered base capable of scavenging protons without acting as a nucleophile. The addition of 0.3 equivalents of DIPEA to individual ammonium groups of CHI-CSA was found to effectively yield graft copolymers after 3 days at room temperature. Upon completion, the products were precipitated with diethyl ether and sequentially washed with THF and water to remove any starting materials and by-products. 

#### 2.1.1. Synthesis of CHI-*graft*-poly(l-lysine(Z))

Synthesis of CHI-graft-poly(l-lysine(Z)) was carried out in standard conditions using l-lysine(Z)-NCA. The ratio between the CHI and l-lysine(Z)-NCA repeating units (CHI/Lys) was estimated by dividing the normalized integral value of the CHI protons 3H–6H (5 protons) by normalized integral value of the l-lysine(Z) B signal (6 protons) (Figure 5). A ratio of 0.28 for CHI to l-lysine was found according to the Equation (2):(2)$$CHI:Lysine=\frac{\frac{1}{5}\times A_{pyr}}{\frac{1}{6}\times B_{lys}}$$
where *A_pyr_* is the area of five protons of the pyranose ring of CHI and *B_lys_* is the area of the six remaining aliphatic protons of l-lysine(Z) (marked B in Figure 5).

The Cbz protecting groups of l-lysine(Z) units were removed in two sequential deprotection reactions using HBr/AcOH and TFA. This yielded a water-soluble product rich in NH_2_ functionalities; the deprotection degree was found to be 61% using Equation (3):(3)$$\%Deprotection=\left[1-\left(\frac{\frac{1}{5}\times A_{ar}}{\frac{1}{6}\times B_{lys}}\right)\right]\times 100$$
where *A_ar_* is the area of the five aromatic protons of the benzyloxycarbonyl protecting group of l-lysine(Z) and *B_lys_* is the area of the six protons of l-lysine (marked B in Figure 5).

#### 2.1.2. Synthesis of CHI-*graft*-poly(l-leucine-co-l-lysine)

Following the established procedure, CHI-*graft*-poly(l-leucine-co-l-lysine) was synthesized by copolymerization of l-leucine-NCA and l-lysine-NCA in the presence of CHI for a 49% yield. A feed ratio of 1:1.5:1.5 (CHI/l-leucine/l-lysine) was selected so that the total CHI/NCA molar ratio remained consistent. As stated previously, l-leucine was selected as the hydrophobic component of the GCP. Additionally, l-leucine allows for straightforward characterization of l-lysine(Z) deprotection as its proton resonances do not overlap with the aromatic protons of the Cbz protecting group. Upon work up, the ratio of CHI to amino acids was determined with ^1^H NMR using Equations (4) and (5): (4)$$CHI:Lys=\frac{\frac{1}{5}A_{pyr}}{\frac{1}{2}A_{lys}}$$
(5)$$CHI:Leu=\frac{{\frac{1}{5}A}_{pyr}}{\frac{1}{3}A_{leu}};A_{leu}=A_{B+C}-6\times (½A_{lys})$$
where *A_leu_* is the area of three protons of l-leucine (marked C in Figure 6) and *A_pyr_* is the area of five protons on the pyranose ring of chitosan. *A_leu_* is calculated by subtraction of *A_lys_*, i.e., the area of two protons of l-lysine bound to the carbons adjacent to the amine group (marked A in Figure 6) from *A_B+C_*, i.e., the total area of the six alkyl protons of l-lysine (marked B in Figure 6) and three alkyl protons of l-leucine (marked C in Figure 6). The CHI/Lys ratio was determined to be 0.35 (Figure 6). With this information the ratio of CHI/Leu can be calculated using the resonance peak of B+C which consists of six l-lysine and three l-leucine protons. Upon substitution of *A_lys_* into Equation (5), the CHI/Leu was found to be 0.22. Similarly, the deprotection degree was found to be 38%.

#### 2.1.3. Synthesis of Block GCPs

Amphiphilic GCPs with block sequences may adopt a number of conformations as a result of various interactions with the local environment [26,27]. Polypeptide block copolymers can be prepared in a one-pot synthesis where the NCAs are sequentially added to the polymerization mixture. This technique allows for the synthesis of polymers with potentially complex structures that are often poorly defined and difficult to characterize. In our work, we synthesized the block graft copolymer CHI-*graft*-poly(l-leucine-block-l-lysine) in a two-step sequential graft copolymerization with l-leucine-NCA and l-lysine-NCA, respectively (Figure 7). First, NCA-ROP of l-leucine initiated by CHI at a ratio of 1:3 (CHI amino groups/NCA) to yield a CHI-*graft*-poly(l-leucine) with a yield of >90%. The product was characterized with ^1^H NMR to determine the actual ratio of CHI:L-leucine. However, the peaks associated with the CHI backbone protons were difficult to observe due to matrix effects. We observed such silencing of CHI in other instances as well. This is due to the fact that the CHI backbone is insoluble in DMSO and the graft copolymer adopts a conformation where CHI collapses while the sidechains swell, thus providing solubility of the entire macromolecule. In the second step, CHI-*graft*-poly(l-leucine) was subjected to NCA-ROP of l-lysine(Z). Ideally, only the amino groups present on the propagating ends of the poly(l-leucine) will initiate polymerization. However, unreacted amino groups on CHI may also serve as grafting sites yielding blocks of poly(l-lysine) directly attached to the CHI backbone. Regardless, the resulting product will still adopt configurations that maximize favorable interactions with the environment and access novel conformations and assemblies. Upon work up, the ratio of the amino acid blocks to CHI was evaluated with ^1^H NMR using Equations (4) and (5). The CHI/Lys ratio was determined to be 0.64 from Equation (4). With this information, the ratio of CHI/Leu was calculated using the combination of the peaks of B and C which consist of six l-lysine and three l-leucine protons. Upon substitution of *A_lys_* into Equation (5), the CHI/Leu ratio was found to be 0.32. Similarly, the extent of Cbz deprotection was determined to be 41% by evaluating the resonances *A_ar_* and *A_lys_* (Figure 8). It is important to note that this value may be underestimated due to the matrix effects. The graft copolymer can adopt conformations that place protected and deprotected l-lysines in distinct environments.

In conjunction with CHI-*graft*-poly(l-leucine-block-l-lysine), which has a hydrophobic core and cationic shell, we made an attempt to synthesize CHI-*graft*-poly(l-lysine-block-l-leucine) “reverse-block”. The polymer was successfully obtained and isolated. Unfortunately, upon deprotection, it was not soluble in any solvent and thus could not be used any further.

FTIR spectroscopic analysis was employed to characterize the final products and confirm the grafting of the peptide chains (Figure 9). CHI was evaluated and compared after each step of the synthesis. In all spectra, the absorbance of amide bands I (1676 cm^−1^), II (1531 cm^−1^), and III (1252 cm^−1^) appeared after ROP, indicating the successful grafting of polypeptides. Additionally, the disappearance of the C-O-C benzyl stretch upon deprotection indicates the removal of the carboxybenzyl group from grafted poly(l-lysine). 

We used SEC to assess the molecular weight of some of the products. In general, characterization of charged polymers with SEC poses substantial challenges caused by specific interactions of the polymer and the stationary phase of the column, and a lack of standards that accurately represent the hydrodynamic radius of nonlinear molecules and polyelectrolytes [28,29]. We used an aprotic solvent DMF in order to minimize ionization of the polymeric products. Only two products were soluble in DMF: CHI-*graft*-poly(l-lysine) and a linear model compound GlcN-*term*-poly(l-lysine). Our attempts to use water as the mobile phase and solvent for SEC resulted in extremely low elution volumes revealing high molecular weights beyond any possibility which indicated either strong ionization of the products or formation of aggregates. 

The results of SEC of CHI-*graft*-poly(l-lysine) and GlcN-*term*-poly(l-lysine) are shown in Figure 10. The molecular weight of GlcN-*term*-poly(l-lysine) is 627 g/mol. Taking into account the weight of the terminal GlcN fragment as 178 g/mol, we determined that the length of poly(l-lysine) is about 3.5 units. This finding fits well with the synthesis when a 3-fold (molar) amount of l-lysine(Z)-NCA monomer was added to GlcN initiator. Some amount of GlcN-*term*-poly(l-lysine) was also observed on the SEC of CHI-*graft*-poly(l-lysine), while the main broad peak averaged 3250 g/mol. Taking into account the molar ratio (0.28) of CHI to l-lysine from the NMR spectrum, we obtained an average of 5.1 poly(l-lysine) chains of an average of 3.5 units in length grafted to a CHI backbone composed of 5.1 units of GlcN. This means that short CHI chains were saturated with poly(l-lysine) sidechains. 

### 2.2. Antimicrobial Activity

Following the characterization of all of our products, we examined the antimicrobial activity of our graft copolymers. AMPs preferentially bind to the plasma membrane bilayer (PMB) of bacteria through electrostatic interactions. The selectivity in binding arises from differences in the relative abundance and distribution of charged and hydrophobic phospholipids. In line with this model, many reports cite that an optimum level of hydrophobicity is required to sufficiently facilitate interactions with fatty acyl chains to trigger membrane permeabilization [30]. However, AMPs with increased hydrophobic content can bind to mammalian membranes and exhibit increased toxicity [31]. 

In the case of CHI-based GCPs, several variables create a more complex picture of selectivity. For example, many of the naturally occurring antimicrobial peptides characterized to date possess a net positive charge, ranging from +2 to +9 [32], whereas the GCPs in this work potentially carry more positive charges per macromolecule. From this perspective, they may bind to cells irrespective of their composition and greatly reduce selectivity or efficiency. Additionally, if the GCPs do in fact deliver a high local concentration of peptide mass to the membrane interface, then off-target interactions with host cells could trigger cell death and lead to increased cytotoxicity. However, membrane adsorption is a poorly defined process in a complex environment. Off-target interactions such as protein adsorption and cation screening may substantially affect the electrostatic potential of the GCP. Additionally, outside the cytoplasmic membrane of both Gram-negative and Gram-positive bacteria, there is a peptidoglycan layer consisting of glycan chains interconnected by peptide side chains [33]. GCPs in this work mimic the composition of bacteria cell walls and may provide complementary interactions in that environment. This additional structural affinity may promote selectivity and offset hydrophobic binding to mammalian cells.

Finally, due to the large molecular weight of the GCPs small changes in the mole fraction of its components can lead to dramatic changes in adsorption behavior. All these traits combined make it difficult to target compositions that exhibit strong selectivity. With this in mind, the synthesized GCPs were initially screened for cytotoxicity. 

The colorimetric MTT assay was utilized to assess mammalian cell biocompatibility and in vitro cytotoxicity. In this experiment the GCPs were tested against Human Dermal Fibroblast (HDF) cells. HDF cells were chosen as they are widely used as a model to mimic the interaction of materials with human skin. They are also key components in inflammatory processes and wound healing. Due to the large molecular weight of GCPs, they are likely best suited for topical applications rather than systemic administration. 

In general, CHI-*graft*-poly(l-lysine) exhibited the lowest toxicity with a minimum cell viability of 65% at 30 mg/mL.

GlcN-*term*-poly(l-lysine) was slightly more toxic with 50% viability at 30 mg/mL. CHI-*graft*-poly(l-lysine-co-l-leucine) and CHI-*graft*-poly(l-leucine-block-l-lysine) exhibited similar profiles and were substantially more toxic with nearly 90% cell death at the solubility limit of 30 mg/mL (See Appendix A). In the case of all four compounds, however, 2.5 mg/mL was found to have little to no effect on HDF viability. Equipped with this information, antibacterial assays of the GCPs were conducted at concentrations below 2.5 mg/mL for therapeutic relevance. The antimicrobial activity of the GCPs against *E. coli* and *S. aureus* was determined using microtiter dilution methods. For each graft copolymer and GlcN-*term*-poly(l-lysine), the minimal inhibitory concentration (MIC) was determined as the lowest concentration of polymer required to inhibit ≥90% of bacteria after overnight incubation. Reductions in the growth of *E. coli* and *S. aureus* vs. polymer concentration are shown in Figure 11 and Figure 12, respectively, and represent the average of at least 12 trials. MIC values were quantified using a nonlinear regression method adapted from Lambert et al. [34] and reported in Table 1. 

The minimum bactericidal concentration (MBC) was determined using AlamarBlue cell viability reagent. When added to bacteria, AlamarBlue is modified by the reducing environment of viable cells and turns red. The wells with the lowest polymer concentration which do not change color correspond to the MBC. The MBC was also confirmed by subculturing wells with the lowest concentration of polymer that inhibited growth onto agar plates. The plates that did not show bacterial growth after overnight incubation corresponds to the MBC (Figure 12).

Our results show that all the polymers can inhibit the in vitro growth of *E. coli* but are largely inactive against *S. aureus* (Table 1). At the highest concentrations above 1 mg/mL, CHI-*graft*-poly(l-lysine) was observed to reduce *E. coli* growth by 50% (Figure 11). Despite possessing hydrophobic content from the CHI backbone, CHI-*graft*-poly(l-lysine) is probably lacking sufficient hydrophobicity to efficiently compromise membrane integrity. CHI-*graft*-poly(l-leucine-block-l-lysine) and CHI-*graft*-poly(l-leucine-co-l-lysine), on the other hand, showed a nearly complete reduction in bacteria growth at similar concentrations. The block-peptide CHI-*graft*-poly(l-leucine-block-l-lysine) was found to be about 30% more potent than the random co-peptide CHI-*graft*-poly(l-leucine-co-l-lysine) despite having similar amino acid compositions. Though the overall activity was similar, the differences were consistent with the hypothesis that block-peptide architectures may facilitate membrane permeabilization more efficiently compared to random co-peptides. However, further study of GCPs with greater contrast between block- and co-peptide composition is needed. Despite these promising results, the GCPs were inactive against *S. aureus* (Figure 12). This is likely due to the presence of a thick peptidoglycan layer on the outer membrane of Gram-positive bacteria which can prevent the diffusion of large molecules.

GlcN-*term*-poly(l-lysine) effectively inhibited the in vitro growth of *E. coli* and *S. aureus* at concentrations well below the MIC of the graft copolymers (Table 1). Similar end-tethered oligo-l-lysine structures synthesized by Singla et al. [35] exhibited comparable antimicrobial activity and were found to induce membrane damage in the tested microbes. Given the biocompatibility and activity of GlcN-*term*-poly(l-lysine), further optimization of such scaffolds could lead to the development of efficient antimicrobial agents. It is important to note that differences in activities of GlcN-*term*-poly(l-lysine) and the graft copolymers may be exaggerated from expressing the MIC in terms of mg/mL rather than µM. Graft copolymers have a substantially larger molecular weight which equates to fewer molecules per gram compared to GlcN-*term*-poly(l-lysine).

MBC was evaluated by subculturing the broth dilution of the MIC test. The Wells above and below the respective MIC were subcultured onto agar plates and incubated for 24 h. As seen in Appendix A, the plates inoculated with GlcN-*term*-poly(l-lysine), CHI-*graft*-poly(l*-*leucine-co-l-lysine), and CHI-*graft*-poly(l-leucine-block-l-lysine) at the respective MIC concentration did not proliferate. However, CHI-*graft-*poly(l-lysine) showed growth at 2.0 mg/mL which agrees with MIC data. 

Another factor potentially influencing the antimicrobial activity of GCPs is the formation of micelles. Several groups have demonstrated effective antimicrobial and antibacterial activity of micelle-forming compounds at concentrations in the nanomolar to low-micromolar range [36]. Amphiphilic polymers form micelles in aqueous solution whereby the polar region faces the outside surface, and the nonpolar region forms the core. In this state, the antimicrobial activity of GCPs may be substantially affected. Additionally, the formation of micelles corresponds to changes in optical properties such as light scattering which may impact absorbance measurements in the MIC assay. In order to determine if the GCPs form micelles at a concentration relevant to the MIC, the optical absorbance was evaluated over increasing concentrations of polymer in MH media (Figure 13). Both GlcN-*term*-poly(l-lysine) and CHI-*graft*-poly(l-lysine) demonstrated non-linear scattering which indicate the formation of micelles. CHI-*graft*-poly(l-leucine-co-l-lysine) and CHI-*graft*-poly(l-leucine-block-l-lysine) on the other hand did not exhibit a change in slope over the relevant concentrations. This observation provides a plausible explanation as to why GlcN-*term*-poly(l-lysine) effectively inhibits in vitro growth of *E. coli* and *S. aureus* at concentrations well below the graft copolymers’ MIC. This hypothesis is a subject of further investigation. 

Biofilms consist of an assembly of microorganisms embedded in a self-produced matrix of extracellular polymeric substances (EPSs) containing polysaccharides, extracellular DNA, proteins, and lipids [37]. They are of clinical relevance due to their ability to colonize medical devices such as catheters and implants. The National Institutes of Health in the USA reported that approximately 80% of chronic infections in humans are biofilm related [38]. 

AMP-mediated strategies for biofilm eradication are an attractive approach that has gained attention in recent years. Compared to traditional small-molecule antibiotics, AMPs offer fast-killing kinetics, a high potential to act on slow-growing or non-growing bacteria, and the ability to synergize with antibiotics [39]. Antimicrobial peptides and polymers have been reported to act at several stages of biofilm development and with different mechanisms of action. Depending on the stage of development, the AMP may inhibit the formation of biofilm or eradicate established biofilms. AMPs that follow an inhibitory pathway typically do so by: (1) altering the adhesion of microbial cells by binding their surfaces or the surface of the substrate, [40] (2) disrupting signaling molecules that regulate biofilm formation [41], or (3) killing early colonizer cells to prevent biofilm maturation [42].

AMPs that target established biofilms follow mechanisms that either kill microbial cells or reduce biofilm mass. Killing pathways typically do so by penetrating the EPS matrix and inhibiting cell division or disrupting the cytoplasmic membrane of microbial cells directly [43]. AMPs that eradicate biofilms reduce film mass by solubilizing components of the EPS matrix via their amphipathic and cationic structures [44].

Here, we study the potential of CHI-based GCPs as an anti-biofilm agent. The synthesized GCPs readily form micelles that may support the solvation of key components in the ECM matrix. Similar materials have been found to act as compatibilizers, stabilizing the solvation of immiscible blends [45,46]. Additionally, the cationic and hydrophobic motifs may promote favorable interaction with the outer membrane of bacteria and neutralize surface charges. Finally, the GCPs may adsorb onto biofouling surfaces before colonization or directly on the biofilm and prevent further development. 

Herein, we evaluated the anti-biofilm activity of the graft copolymers series against *A. tumefaciens. A. tumefaciens* was selected as a model bacterium as it readily forms biofilms, is generally safe to humans, and has well-established assays and protocols for evaluation.

The adherent biofilm mass after treatment with the respective graft copolymers is shown in Figure 14. When compared to untreated bacterial cells, both GlcN-*term*-poly(l-lysine) and CHI-*graft*-poly(l-leucine-co-l-lysine) showed significant reductions in the adherent mass of WT and ΔvisR strains. CHI-*graft*-poly(l-leucine-block-l-lysine), on the other hand, increased biofilm mass of wild type while decreasing ΔvisR. No changes were observed in the biofilms of the Δupp strain for any of the graft copolymers. Given that Δupp lacks the exopolysaccharide adhesions that are critical for surface attachment, exposure to the graft copolymers could only result in increased adherent mass.

Differences in the activity of CHI-*graft*-poly(l-leucine-block-l-lysine) when exposed to wild type or ΔvisR were unexpected. Some antimicrobial peptides are known to target a wide range of intracellular components, including DNA, RNA, and proteins [47]. Binding to any of these sites is complex as it can lead to opposing trends in biofilm regulation. Subtle changes in gene expression or the intracellular composition of ΔvisR may facilitate graft copolymer binding and alter biofilm expression.

The direct antimicrobial effects of the graft copolymers on planktonic *A. tumefaciens* were also evaluated. Optical density measurements of the biofilm inoculum were taken before and after film formation (Figure 15). Reductions of 50% or greater were observed for all graft polymers and strains of bacteria at a concentration of 0.25 mg/mL. This implies that direct antimicrobial activity of the GCPs in solution may contribute to biofilm losses. 

The extent to which reductions in biofilm mass correspond to reductions in planktonic bacteria was evaluated by normalizing the ratio of biofilm mass to the number of planktonic bacteria (Figure 16). Values greater than 100% correspond to elevated biofilm formation, while values less than 100% indicate the biofilm was disproportionately reduced relative to viable planktonic bacteria. 

Wild type *A. tumefaciens* treated with glucosamine-terminated poly(lysine) exhibited reductions in biofilm mass and planktonic bacteria that were in proportion to untreated bacteria. This suggests the anti-biofilm activity of GlcN-*term*-poly(l-lysine) is a result of antimicrobial action rather than selective inhibition. CHI-*graft*-poly(l-leucine-co-l-lysine), on the other hand, reduced adherent mass without killing planktonic bacteria. Finally, CHI-*graft*-poly(l-leucine-block-l-lysine) was found to increase adherent mass despite reducing viable bacteria. Further investigation is required to understand the mechanism of this paradoxical behavior. 

## 3. Materials and Methods

### 3.1. Materials

Chitosan of low molecular weight (Brookfield viscosity 20–300 cP, for 1 wt.% in 1% acetic acid; Mv = 50–190 kDa; 90% deacetylated), 10-camphorsulfonic acid (β) (CSA, 98%), *N*,*N*-diisopropylethylamine (DIPEA, 99.5%), d-(+)-glucosamine hydrochloride (GlcN, >99%), *N*-acetyl-d-glucosamine, (GlcNAc, >98%), and 3-nitrobenzonitrile (3-NBN, 98%) were purchased from Millipore Sigma (Burlington, Burlington Township, NJ, USA) and used as is. *N*-ε-carbobenzyloxy-l-lysine (l-lysine(Z)-OH, 98%) and triphosgene (>99%) were purchased from Oakwood Chemical Inc. (Estill, SC, USA). (4*S*)-4-(2-methylpropyl)-1, 3-oxazolidine-2, 5-dione (l-leucine-NCA) of 98% purity was purchased from OXCHEM corporation. Dimethyl sulfoxide (DMSO) and tetrahydrofuran (THF) were dried over 3Å molecular sieves to remove water and stored under argon until use. All other solvents were of analytical grade and used without further purification. Glassware used for NCA synthesis and polymerization were dried at 120 °C under vacuum overnight.

Magainin and Müeller–Hinton media/agar were purchased from Sigma Aldrich (St. Louis, MI, USA). AlamarBlue™ Cell Viability Reagent and MTT (3-(4,5-Dimethylthiazol-2-yl)-2,5-Diphenyltetrazolium Bromide) were purchased from ThermoFisher Scientific (Waltham, MA, USA). All other materials and supplies were purchased from MilliporeSigma.

The following established bacterial strains were used in this study, all of which are included in our strain collection: *A. tumefaciens* strain C58 and its isogenic deletion derivatives lacking either *visR* or the *upp* loci [48,49], *Escherichia coli* strain BW21553 (Coli Genetic Stock Center), and *Staphylococcus aureus* strain F-182 (ATCC 43300).

### 3.2. Methods

#### 3.2.1. Synthesis of CHI-CSA Salt 

Using a method adapted from Sashiwa et al. [19], 10-camphorsulfonic acid salt of chitosan (CHI-CSA) was prepared as follows. CHI (5.0 g, 30 mmol of NH_2_) was suspended in 1.0 L of water and 10-camphorsulfonic acid (CSA) (7.3 g) in an equimolar amount to NH_2_ of CHI was added. The suspension was stirred until a clear solution formed (approx. 1 h). The solution was coarsely filtered and dialyzed in a cellulose membrane (MW cut-off 12,000 Da) against Milli-pure water for 3 days. The solution was then lyophilized to a fluffy white solid and stored under argon. 

#### 3.2.2. Synthesis of l-lysine(Z)-NCA

l-lysine(Z)-OH (1.5 g, 5.3 mmol) was dried under vacuum and added to 50 mL of anhydrous ethyl acetate in a dried round bottom flask under argon. Triphosgene (0.8 g, 2.7 mmol) was added to the suspension and the mixture was heated to 60 °C with continuous stirring for 3 h. Upon the formation of a clear solution, the reaction was cooled to 0 °C and extracted with water until a neutral pH of the aqueous phase was achieved. The purified organic phase was dried with MgSO_4_ and the solvent was removed under vacuum. The resulting white solid was recrystallized from THF/hexane (1:3) to remove any residual triphosgene and dried under high vacuum overnight to obtain l-lysine(Z)-NCA. Yield: 1.6 g, (75%).

#### 3.2.3. Synthesis of CHI-*graft*-poly(l-lysine(Z))

In a dry 250 mL round bottom flask, CHI-CSA (0.8 g, 2.0 mmol) was dissolved in 125 mL of anhydrous DMSO and equipped with a stir bar. Once dissolved, a three-fold excess l-lysine(Z)-NCA (1.87 g, 6.1 mmol) was added to the solution. The reaction proceeded at 23 °C while stirring over a period of 5 days. CHI-*graft*-poly(l-lysine(Z)) polymer solution was isolated by precipitation in diethyl ether (3 × 100mL) and collected by centrifugation. The resulting residue was dried under vacuum to yield a white solid, 1.78 g (74%).

#### 3.2.4. Synthesis of CHI-*graft*-poly(l-leucine-co-l-lysine(Z))

Synthesis was carried out in a similar fashion to CHI-*graft*-poly(lysine(Z)). In a dry 250 mL round bottom flask, CHI-CSA (0.8 g, 2.0 mmol) was dissolved in 125 mL of anhydrous DMSO and equipped with a stir bar. Once dissolved, l-lysine(Z)-NCA (0.937 g, 3.0 mmol) and l-leucine-NCA (0.479 g, 3.0 mmol) were added to the solution in a 1.5-fold molar excess to CHI amines. The reaction proceeded over 5 days and the final product was isolated by precipitation in diethyl ether and collected by centrifugation. The resulting residue was dried under vacuum to yield a white solid, 0.74 g (49%). 

#### 3.2.5. Synthesis of CHI-*graft*-poly(l-leucine-block-lysine(Z))

Using the method described above, CHI-*graft*-poly(l-leucine) was first synthesized. In a dry round bottom flask, 0.69 g of CHI-*graft*-Poly(leucine) was dissolved in 125 mL of anhydrous DMSO and equipped with a stir bar. Once dissolved, l-lysine(Z)-NCA (1.64 g, 5.3 mmol) was added to the solution. The reaction proceeded at 23 °C while stirring over a period of 5 days. The product was isolated by precipitation in diethyl ether and collected by centrifugation. The resulting residue was dried under vacuum to yield a white solid, 0.76 g (55%). 

#### 3.2.6. Synthesis of Linear GlcN-*term*-poly(l-lysine(Z))

In a dry 250 mL round bottom flask, 1.5 mmol of GlcN was dissolved in 125 mL of anhydrous DMSO and equipped with a stir bar. Once dissolved, a 3-fold excess of NCA (4.5 mmol) was added to the solution. The reaction proceeded at 23 °C while stirring over a period of 5 days. The final products were isolated by precipitation in diethyl ether and collected by centrifugations residue was dried under high vacuum overnight to yield off-white solids.

#### 3.2.7. Deprotection of N-ε-Carbobenzyloxy-l-lysine (l-lysine(Z)) Moieties

*N*-ε-Carbobenzyloxy-l-lysine products were deprotected by HBr. Generally, 1.0 g of polymer was dissolved in 15 mL of trifluoroacetic acid at 0 °C in a sealed vessel. Once dissolved, 15mL of HBr (33% in acetic acid) was added and stirred for 45 min. The polymer was precipitated with diethyl ether (150 mL) and then collected by centrifugation. The final products were washed two additional times with diethyl ether and dried under high vacuum to yield an off-white solid. 

### 3.3. Characterization 

Size exclusion chromatography (SEC) was used to characterize the molecular weight of the GCPs and model linear polypeptides GlcN-*term*-poly(L-lysine). SEC analysis was performed using a Perkin-Elmer Series 200 HPLC/SEC with a Waters 410 RI detector. The column used was GRAM (50 mm × 8 mm, particle size 10 μm) from PSS Polymer Standards Service GmbH (Mainz, Germany). DMF was used as a mobile phase for analysis. The column was calibrated with 5 polystyrene standards in the MW range of 600 to 100,000 Da. GCPs and model linear polypeptide (1% solutions in DMF) were filtered through a 0.2 µm filter and used for SEC. The flow rate for analysis was 0.5 mL/min at 40 °C, and the injection volume was 40 µL.

1H-NMR spectroscopy was performed at room temperature using a Bruker (Billerica, MA, USA) Advance III (400 MHz) NMR spectrometer and analyzed with Bruker’s TopSpin software. Spectra of the products were evaluated against precursors with DMSO-d_6_ as a reference. 

IR spectra were collected using a Nicolet iS10 FTIR spectrophotometer (Waltham, MA, USA) with a Smart iTX Diamond ATR accessory from 4000 cm^−1^ to 400 cm^−1^. Each experiment used 32 scans.

#### 3.3.1. Minimum Inhibitory Concentrations (MICs)

Bacteria cells were grown overnight at 37 °C in Müeller–Hinton (MH) media to a mid-log phase and diluted to 10^4^ to 10^5^ CFU mL^−1^ in media. A twofold dilution series of a 100 μL drug solution in the MH media was made in a 96-well microplate, followed by the addition of 5.0 μL bacterial suspension (10^4^ to 10^5^ CFU mL^−1^). The plates were incubated at 37 °C for 18 h, and the absorbance at 600 nm was measured with a microplate reader (BioTek Synergy HT, Winooski, VT, USA). Additionally, 10 μL of AlamarBlue Cell viability reagent assay was added to each well and incubated at 37 °C for an additional 2 h. The absorbance at 570 nm was measured. Positive control measurements were performed without product, and the negative control was wells without bacteria/inoculum. MICs were determined as the lowest concentration that inhibited cell growth by ≥90% using curve-fitting.

#### 3.3.2. LIVE/DEAD Assay to Examine Bacterial Viability

The number of viable cells after exposure to test compounds was confirmed by serially diluting aliquots of bacteria in Müeller–Hinton media and plating onto Müller–Hinton Agar for *A. tumefaciens*. The plates were incubated overnight and the numbers of live bacteria were enumerated and expressed as CFU mL^−1^. Experiments were performed in triplicate.

#### 3.3.3. Crystal Violet Biofilm Assay 

Static-culture biofilms were grown on sterile PVC coverslips suspended vertically in the wells of UV-sterilized 12-well polystyrene dishes. Overnight cultures were diluted to an initial optical density at 600 nm (OD_600_) of 0.05 in ATGN media. Each well was inoculated with 3.0 mL of culture, and the dishes were incubated at room temperature for 12 to 96 h. Coverslips were stained with 0.1% crystal violet (CV) dye. Quantification of the adherent biomass was achieved via solubilization of adsorbed CV with 33% acetic acid. The OD_600_ of the 48 h planktonic cultures and the *A*_600_ of the solubilized adherent CV were measured with a BioTek Synergy HT plate reader. The values shown are the mean A_600_ +/- standard error of three biofilm cultures for each strain/condition.

#### 3.3.4. Test of Activity against Biofilms

The anti-biofilm activity of the GCPs was evaluated by inoculating wild type *A. tumefaciens* and the mutants ΔvisR and Δupp before colonizing the substrate surface. The deletion mutants ΔvisR and Δupp were included as controls which exhibit increased and decreased biofilm production, respectively. Changes in biofilm mass were quantified using a static biofilm coverslip assay optimized for *A. tumefaciens* [18]. Briefly, the agrobacterium strains were cultured overnight and diluted to an optical density of 0.05 A.U. in 35 mm wells. The cultures were inoculated to a concentration of 0.25 mg/mL with the respective GCP and a polyvinyl chloride coverslip was suspended in the solution. After a 24 h incubation, planktonic bacteria growth was quantified by spectrophotometry and the colonized coverslips were stained with crystal violet solution. Images of the stained biofilms were collected for qualitative analysis. Finally, crystal violet was solubilized from the films and the absorbance was measured at 600 nm to estimate the relative amounts of adhered biomass. 

## 4. Conclusions

In conclusion, this work describes the synthesis and characterization of cationic CHI-based GCPs. Utilizing a “grafting from” approach, the hydrophobic and cationic amino acids were conjugated to the polysaccharide CHI via ring-opening polymerization of NCA derivatives of two amino acids: l-lysine and l-leucine. The polymers were designed as a novel antimicrobial agent that adheres to traditional structure–function relationships of AMPs while also mimicking the peptidoglycan structure of bacteria. GCPs with block-peptide chains were successfully synthesized in a two-step sequential synthesis. Several amino acid combinations and sequences were synthesized. However, CHI-*graft*-poly(l-lysine), CHI-*graft*-poly(l-leucine-co-l-lysine), and CHI-*graft*-poly(l-leucine-*block*-l-lysine) were optimal for further study. The structure of the GCPs was evaluated using NMR, FTIR, and SEC.

Additionally, the antimicrobial activity and biocompatibility of cationic GCPs were evaluated. Compared to linear antimicrobial polymers, GCPs have greater conformational freedom and charge density which may lead to enhanced bactericidal activity and unique modes of action. Additionally, GCPs with a polysaccharide backbone may have favorable interactions with the peptidoglycan layer present in bacteria cell walls. A small set of representative polymers, CHI-*graft*-poly(l-lysine), CHI-*graft*-poly(l-leucine-co-l-Lysine), and CHI-*graft*-poly(l-leucine-*block*-l-lysine), along with glucosamine-terminated GlcN-*term*-poly(l-lysine) were selected for the study. The in vitro cytotoxicity of the GCPs was evaluated against human dermal fibroblasts (HDFs). Of the tested compounds, GlcN-*term*-poly(l-lysine) and CHI-*graft-*p(l-lysine) exhibited the lowest toxicity. All four polymers demonstrated good cytocompatibility at 2.5 mg/mL. All the polymers were found to inhibit in vitro growth of *E. coli* but were largely inactive against *S. aureus.* CHI-*graft*-poly(l-leucine-co-l-lysine) was found to be twice as potent as CHI-*graft*-poly(l-leucine-block-l-lysine). GlcN-*term*-poly(l-lysine) effectively inhibited in vitro growth of *E. coli* and *S. aureus* at concentrations well below MIC values. The anti-biofilm activity of the GCPs was evaluated against *A. tumefaciens*, a plant pathogen of agricultural relevance. Both GlcN-*term*-poly(l-lysine) and CHI-*graft*-poly(l-leucine-co-l-lysine) showed significant reductions in the adherent mass of WT and Δ*visR* strains. On the other hand, CHI-*graft*-poly(l-leucine-block-l-lysine) increased the biofilm mass of wild type while decreasing Δ*visR*. The extent to which the reductions in biofilm mass correspond to reductions in planktonic bacteria was evaluated by normalizing the ratio of biofilm mass to the number of planktonic bacteria. GlcN-*term*-poly(l-lysine) reduced adherent mass proportionally to cell viability whereas CHI-*graft*-poly(l-leucine-co-l-lysine) more selectively inhibited biofilm formation. Finally, CHI-*graft*-poly(l-leucine-block-l-lysine) was found to reduce planktonic cell viability while increasing adherent biofilm mass. 

## Figures and Tables

**Figure 1 marinedrugs-21-00243-f001:**
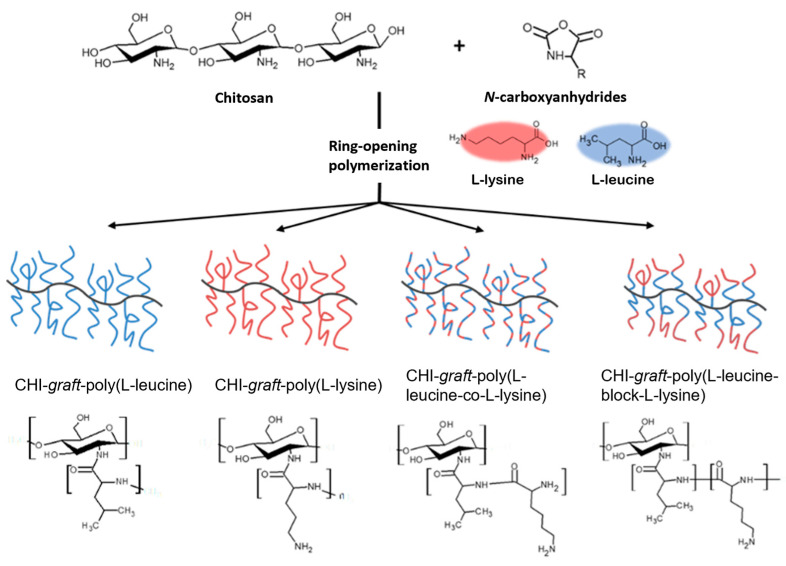
Synthetic methodology developed to fabricate cationic peptidopolysaccharide graft copolymers consisting of L-lysine and L-leucine oligomers conjugated to chitosan via ring-opening polymerization of *N*-carboxyanhydrides (NCA-ROP).

**Figure 2 marinedrugs-21-00243-f002:**
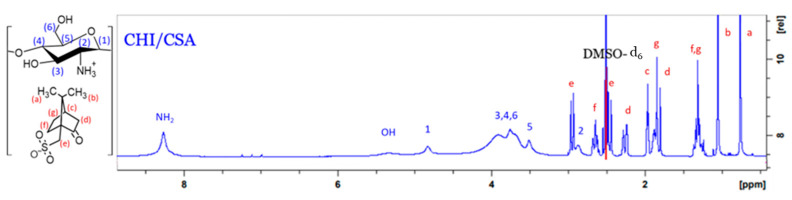
^1^H-NMR spectrum of the CHI-CSA in DMSO-d_6_ with schematic presentation of repeat unit.

**Figure 3 marinedrugs-21-00243-f003:**
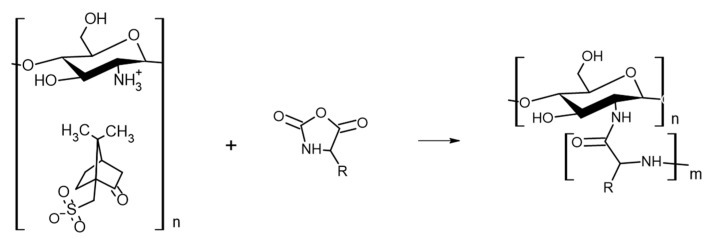
NCA-ROP synthesis of CHI-based GCPs.

**Figure 4 marinedrugs-21-00243-f004:**
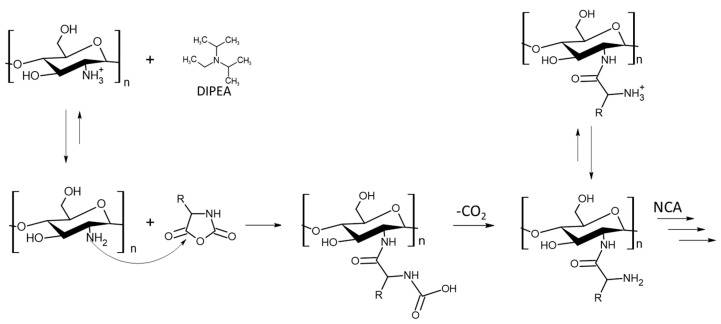
Tentative mechanism of the NCA-ROP using CHI-CSA as macroinitiator.

**Figure 5 marinedrugs-21-00243-f005:**
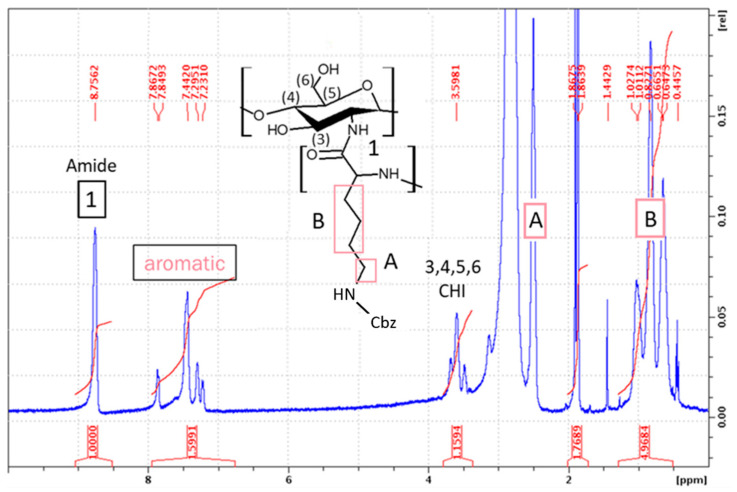
^1^H NMR spectra (DMSO-d_6_) of CHI-*graft*-poly(l-lysine(Z)).

**Figure 6 marinedrugs-21-00243-f006:**
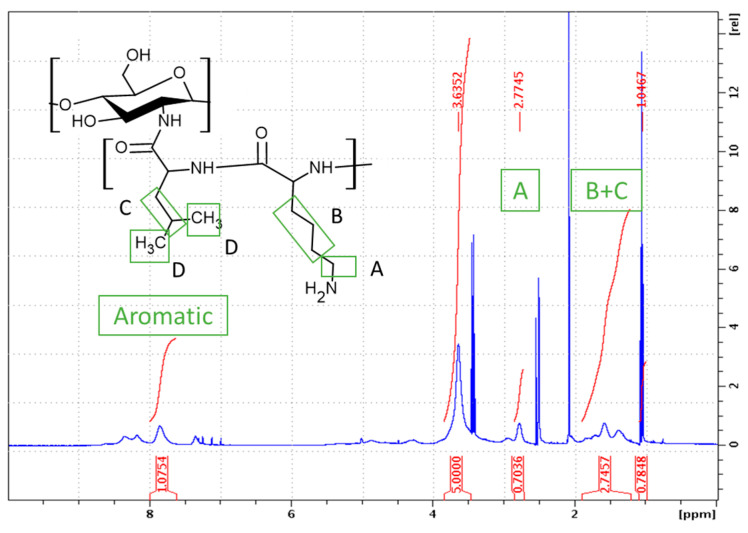
^1^H NMR spectra (DMSO-d_6_) of CHI-*graft*-poly (l-leucine-co-l-lysine).

**Figure 7 marinedrugs-21-00243-f007:**
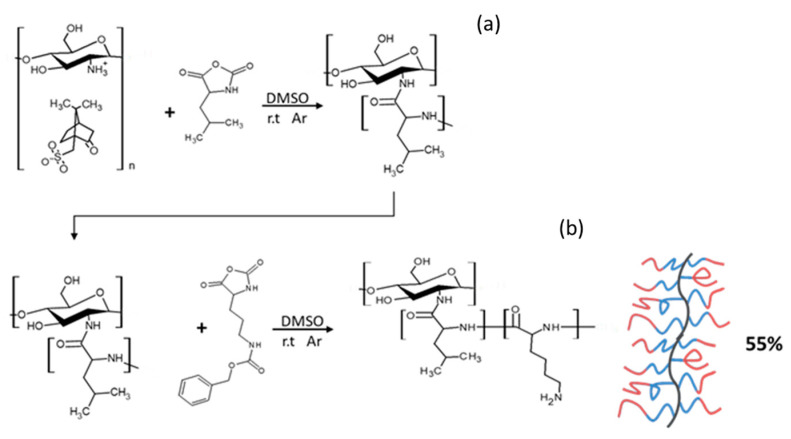
Synthesis of CHI-*graft*-poly(L-leucine-block-L-lysine) (“Block”) via sequential ring-opening polymerization of L-leucine (**a**) and L-lysine(Z) (**b**) in a two-step process.

**Figure 8 marinedrugs-21-00243-f008:**
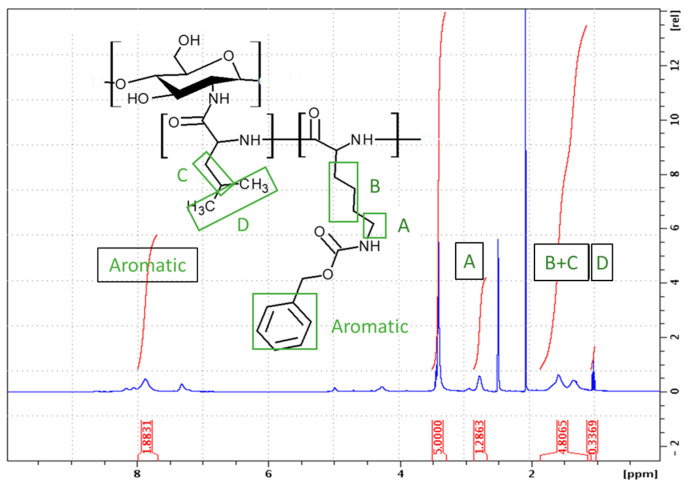
^1^H NMR spectra (DMSO-d_6_) of CHI-*graft*-poly(l-leucine-block-l-lysine).

**Figure 9 marinedrugs-21-00243-f009:**
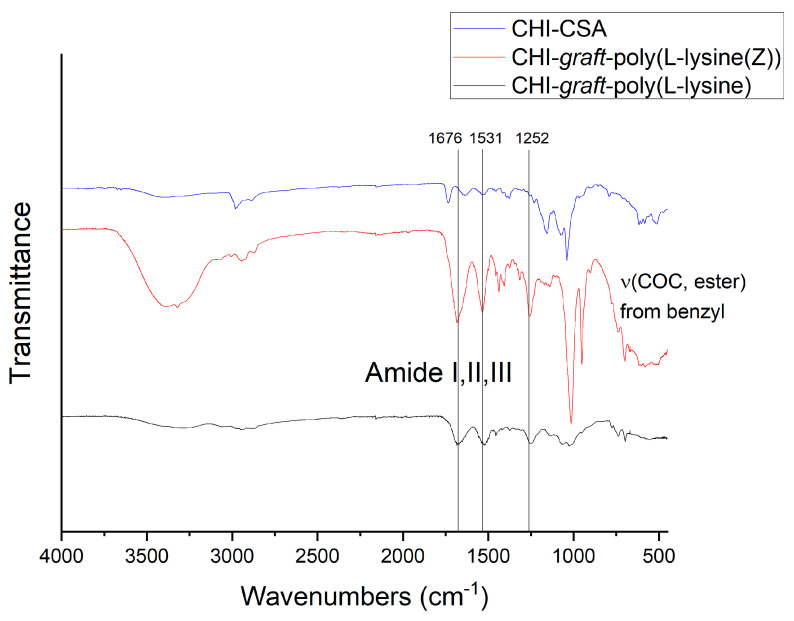
Representative FTIR spectra of CHI-CSA (blue), CHI-*graft*-poly(L-lysine(Z)) (red), and CHI-*graft*-poly(l-lysine) (black).

**Figure 10 marinedrugs-21-00243-f010:**
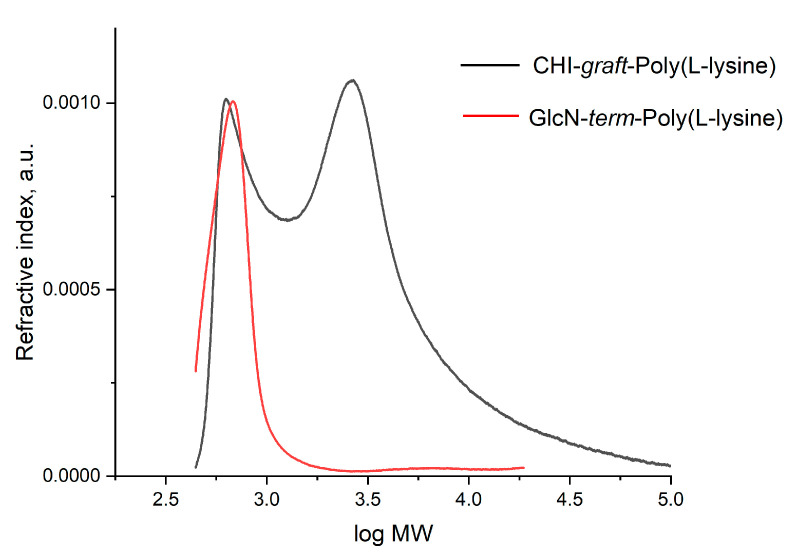
SEC of CHI-*graft*-poly(L-lysine) (black) and a linear model compound GlcN-*term*-poly(L-lysine) (red) in DMF.

**Figure 11 marinedrugs-21-00243-f011:**
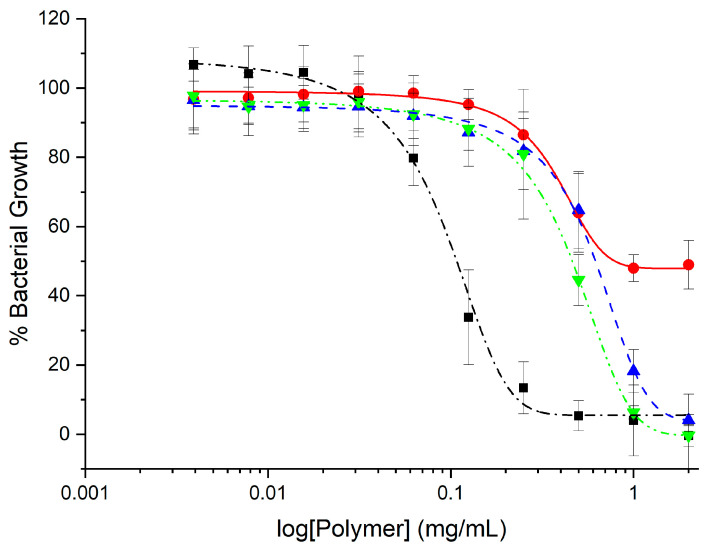
Growth inhibition curve of *E. coli* treated with different concentrations of GlcN-*term*-poly(l-lysine) (black squares), CHI-*graft*-poly(l-lysine) (red circles), CHI-*graft*-poly(l-leucine-co-l-lysine) (blue triangles), and CHI-*graft*-poly(l-leucine-block-l-lysine) (green triangles) for 24 h. Data are presented as the mean and standard deviation, *n* = 12. The half maximal inhibitory concentrations, IC_50_, of tested GCPs estimated from these data are as follows: GlcN-*term*-poly(l-lysine—0.10 mg/mL, CHI-*graft*-poly(l-leucine-block-l-lysine)—0.51 mg/mL, CHI-*graft*-poly(l-leucine-co-l-lysine)—0.66 mg/mL, and CHI-*graft*-poly(l-lysine)—0.85 mg/mL.

**Figure 12 marinedrugs-21-00243-f012:**
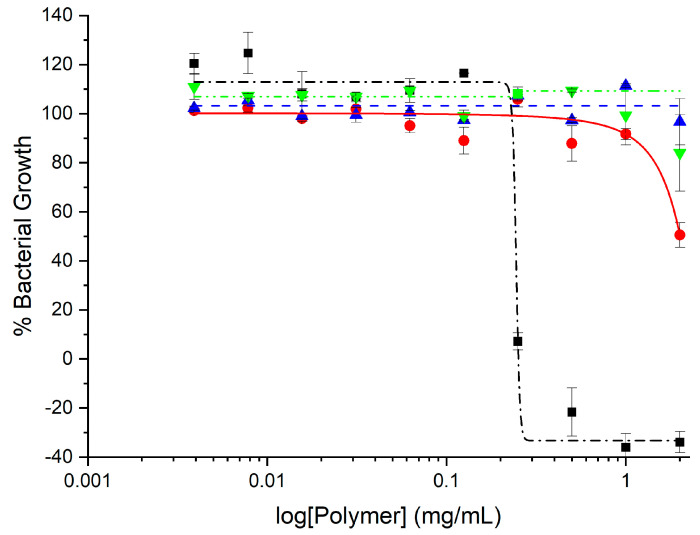
Growth inhibition curve of *S. aureus* treated with different concentrations of GlcN-*term*-poly(l-lysine) (black squares), CHI-*graft*-poly(l-lysine) (red circles), CHI-*graft*-poly(l-leucine-co-l-lysine) (blue triangles), and CHI-*graft*-poly(l-leucine-block-l-lysine) (green triangles) for 24 h. Data are presented as the mean and standard deviation, *n* = 12.

**Figure 13 marinedrugs-21-00243-f013:**
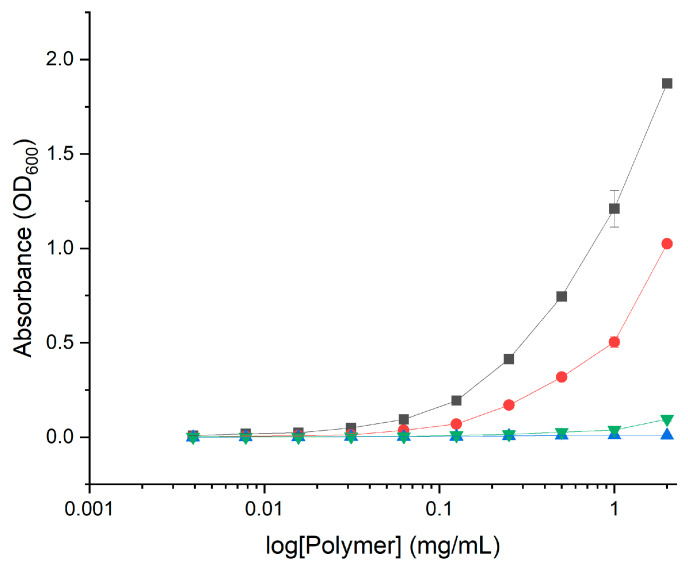
Absorbance of the graft copolymers at varying concentrations in MH media at 600 nm: GlcN-*term*-poly(l-lysine) (black squares), CHI-*graft*-poly(l-lysine) (red circles), CHI-*graft*-poly(l-leucine-co-l-lysine) (blue triangles), and CHI-*graft*-poly(l-leucine-block-l-lysine) (green triangles) for 24h. Data are presented as the mean ± standard deviation, *n* = 4.

**Figure 14 marinedrugs-21-00243-f014:**
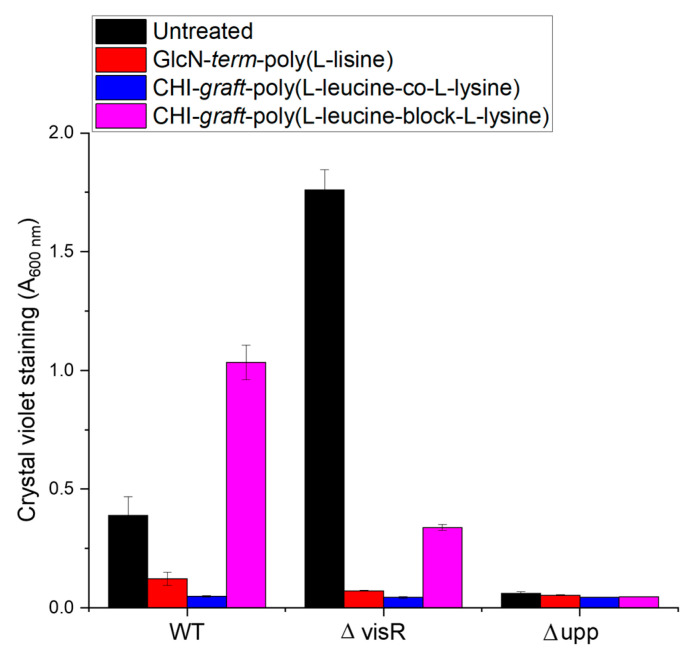
Inhibition of adherent biomass upon exposure to CHI-based GCPs. *A. tumefaciens* was grown in the presence of the respective graft copolymers at a concentration of 0.25 mg/mL. The ΔvisR and Δupp mutants exhibited enhanced and depleted film formation, respectively. Total adherent biomass was quantified by crystal violet staining. Mean values of three independent experiments and standard error are shown.

**Figure 15 marinedrugs-21-00243-f015:**
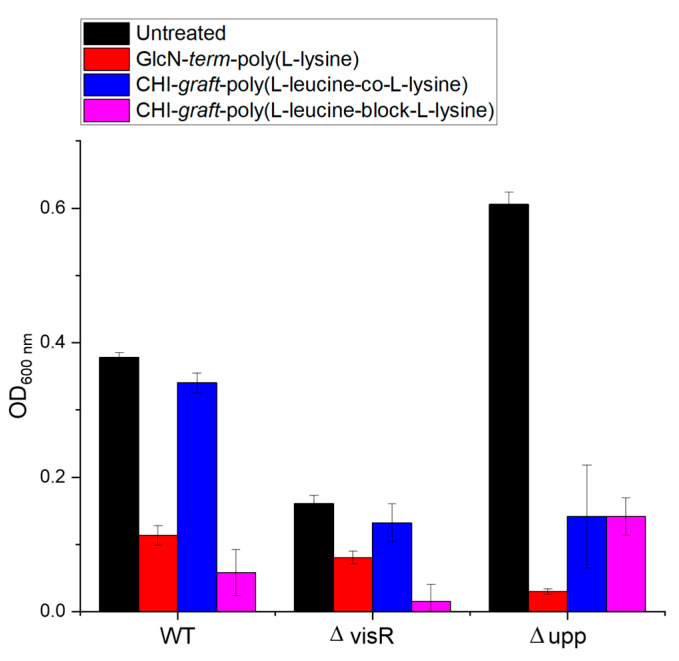
Optical density of *A. tumefaciens* planktonic biomass treated with 0.25 mg/mL of each polymer. Values are averages of triplicate assays and error bars represent standard deviation.

**Figure 16 marinedrugs-21-00243-f016:**
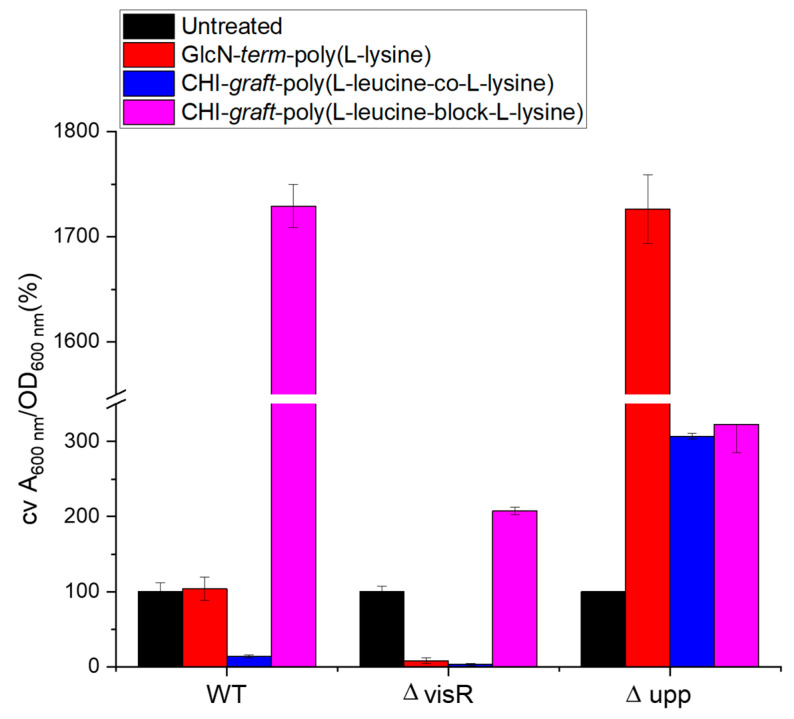
Changes in adherent biofilm mass (A_600_) of *A. tumefaciens* normalized by planktonic biomass (OD_600_) treated with 0.25 mg/mL of each polymer. Values are averages of triplicate assays and error bars represent standard deviation. Data for each strain are normalized (100%) to non-treated cultures of each strain (black bars).

**Table 1 marinedrugs-21-00243-t001:** Minimum inhibitory concentrations (MICs) and minimum bactericidal concentrations (MBCs) of the synthesized polymers, mg/mL. The values are the average of three replicates.

Polymer	*E. coli*		*S. aureus*	
	MIC	MBC	MIC	MBC
GlcN-term-poly(l-lysine)	0.16 ± 0.09	0.12	0.26 ± 0.12	0.25
CHI-graft-poly(l-lysine)	0.65 ± 0.11	>2.0	>2.0	>2.0
CHI-graft-poly(l-leucine-co-l-lysine)	1.2 ± 0.02	2.0	>2.0	>2.0
CHI-graft-poly(l-leucine-block-l-lysine)	0.76 ± 0.04	1.0	>2.0	>2.0
Magainin II	0.04 ± 0.03	0.06	Not tested	

## Data Availability

All data generated or analyzed during this study are included in this published article.

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
