# Peer review of "Synthesis and Antibiotic Activity of Chitosan-Based Comb-like Co-Polypeptides"

_marinedrugs, 2023, doi:10.3390/md21040243_

Round 1

Author Response

We deeply appreciate the diligent revision made by the reviewer 1. All comments are addressed. We made all small corrections (typos, format issues, grammatical faults) suggested by the reviewer: #4-23, #25-38, #40-51, #53-57, #59, #60, #62, #65, #67-69, and #71-73.

#1: All polymers have been uniformly and coded and the consistence has been triple-checked. We apologies for the inconsistency in the original manuscript.  

#2, 3. The abstract has been edited to fix grammatic faults.

#24. These schematics are important part of the discussion on the NCA synthesis, ROP, and grafting from CHI. Therefore, we believe they are not a part of M&M, but important illustration of discussion of our synthetic work.

#39. The quality and informativity of Figure 10 (SEC) has been improved. In addition, more diligent description of the SEC experiment including corrections of the detector used (RI instead of UV) and details about calibration has been added to M&M part.

#52. Corrected. It is about 30% more potent. Also please note that the X axis is in logarithmic scale. It visually makes the difference in concentration less expressed.

#58. We prefer to leave the standard deviation values to demonstrate that our results are fairly reproducible.

#61. We edited the caption for Figure 13. We hope you will find them clear. Also, we moved the figure to SI as you recommended. The value in the Table 1 is corrected.

#63. Thank you. The reference on Zhou et al (2016) was added. Also, we added a comment on this correlation in the text, L 494-497.

#64. The section on the anti-biofilm studies was shortened. Yet, we feel that complete removal of this section will leave too many questions. We also would like to avoid moving it to the introduction as it does not logically fit there.

#66. We have moved this section to M&M; it is now “2.3.4. Test of activity against biofilms.” Thank you for this suggestion.

#70. The MS part has been removed. Unfortunately, our MS data showed no consistency and could not be interpreted.

Once again, we are thankful for efforts and time of the reviewer and believe that the new version of the manuscript is substantially improved.

Reviewer 2 Report

The authors in the paper titled: “Synthesis and antibiotic activity of Chitosan-based comb-like co-polypeptides” described the synthesis of Chitosan-based polypeptides and their activity against clinically significant pathogens and the disruption of biofilm formation. The paper is in general well written but it has many typos.

Typos:

Line 258  Instead of “bi-products” it should be by-products.

Line 359  Instead of “corresponding” it should be corresponding.

Line 451, 452, 647, 648  Chi- should be CHI-

L- in L-Leu, and L-Lys should be written in small caps.

Line 420, 625, 632 Instead of GLU it should be GlcN

The abbreviation should be described with the first occurrence not like with Agrobacterium tumefacines full name was mentioned in Conclusion line 627.

Some bigger mistakes:

Formula 2 should be written correctly, in correlation with the description.

Formula 5 needs better explanation.

Figure 8 shows the specter of CHI-graft-poly(L-lysine-block-L-leucine), which is not used in further investigation, while the specter of CHI-graft-poly(L-leucine-block-L-lysine) is not shown.

In further text very often these two (CHI-g-p(L-Lys-block-L-Leu and  CHI-g-p(L-Leu-block-L-Lys)) compounds were mixed, and it isn't easy to understand which compound the authors are talking about.

When an abbreviation is used for these compounds, there is no need for the full name in the following text. A numbering of compounds can also be used.

The manuscript can be published in Marine Drugs after a complete text check.

Author Response

We appreciate the revision made by the reviewer 2. All comments are addressed. We made all small corrections (typos, format issues, grammatical faults) suggested by the reviewer.

Also:

Equations 2 and 5 have been revised and simplified, uniform coding is used for clarity.

Figure 8 and the related text have been edited. CHI-graft-poly(L-leucine-block-L-lysine) and CHI-graft-poly(L-leucine-co-L-lysine) were only two copolymers investigated in this work. In conjunction with CHI-graft-poly(L-leucine-block-L-lysine), which has a hydrophobic core and cationic shell, we made an attempt to synthesize CHI-graft-poly(L-lysine-block-L-leucine) “reverse-block”. The polymer has been successfully obtained and isolated. Unfortunately, upon deprotection it was not soluble in any solvent and thus could not be used any further.

We apologize for the confusion caused by inconsistent coding of the polymers in original manuscript. In the current manuscript all polymers are uniformly coded. The consistence of the coding has been triple-checked. 

Reviewer 3 Report

The authors carry out the synthesis of chitosan copolymers with lysine and leucine in different types of arrangements. They carry out antimicrobial tests, with model bacteria (E. coli and S. aureus), and with A. tumefaciens as a biofilm former.

The article is very interesting, and the results obtained are quite encouraging in the search for new alternatives to antibiotics.

I think the article is well written and I only have some minor comments:

My area of expertise is not organic synthesis, so considering the results and analyzes performed, I assume that the desired products were obtained.

Regarding the results with the free bacteria, a plausible explanation or a possible mechanism of action of the copolymers is not offered. Cationic AMPs generally act by disrupting the membrane, forming pores by different mechanisms. What mechanism could the authors propose for the activity of these polymers? taking into account that its molecular weight is quite higher than an AMP. Perhaps some tests could be done to verify the disruption of the membrane, for example with sytox green (doi), which could enrich the results.

The results show that the best field of application can be aimed at biofilm formation. In this area, the results are very interesting, and in particular the differences found with CHI_lysblockleu, which increases biofilm formation, despite killing planktonic bacteria. In this regard, is it not possible that the matrix is being "enriched" by the copolymer? Is it not possible to differentiate between them in the tests?

Minor issues

the GlcN-term-poly(L-lysine), as it is coded at the beginning of the article and in the figures, changes in the last paragraphs to GLU-term-p(lys), please unify the terms

Review the spaces between the amounts and the units for example

L96    5g

L113 2.4g

L137,138 0.8g, 0.937g.

L161 15mL.  L497, L499, L547, L549, L552

Please use italic fonts for organism names such as E. coli, S. aureus, A. tumefaciens and Latin words such as in vitro

Some typos

L325 combination instead of combination

L358 compounds instead of compunds

L359 corresponding instead of corresponding

L412 terminated instead of terminiated

L418 lack of space therapeutic relevance.The ….

Author Response

We appreciate the revision made by the reviewer 3. All comments are addressed. We made all corrections (typos, format issues, grammatical faults) and formatting suggested by the reviewer. Also, in the edited manuscript all polymers have been uniformly coded and formatted. The consistence has been triple-checked. We apologies for the inconsistency of the original manuscript.   

Regarding the results with the free bacteria. Yes, we believe that GCPs may disintegrate the cell membranes in a manner similar to AMPs, in spite of relatively high molecular mass. Micelle formation of GCPs may promote this process. We added some consideration in the text of edited manuscript,  L 481-496. 

Regarding biofilm formation. The idea of ECM enrichment with the copolymer is very interesting, thank you. We believe that CHI may play a signifcant role in penetration into and accumulation in the ECM of the biofilms. This hypothesis is a subject of further investigation when CHI backbone (polysaccharide) will be substituted with a different polymer (non-polysaccharide).

Round 2

Reviewer 1 Report

R&D (L454-L455): The authors state: The random co-peptide CHI-graft-poly(L-leucine-co-L-lysine), was found to be about 30% more potent than the block-peptide CHI-graft-poly(L-leucine-block-L-lysine)”. According to Figure 11, it seem to be the opposite. CHI-graft-poly(L-leucine-block-L-lysine) (green triangles/green line) seem to be more potent than CHI-graft-poly(L-leucine-co-L-lysine) (blue triangles (blue line). The more to the left the curve is, the lower concentration is needed to inhibit bacterial growth (lower IC50-values), i.e. the more potent the polymer is.

R&D (L509): “in vitro” should be in Italic.

R&D (L530-531): I suggest changing “Depending on the stage of development the AMP targets it may inhibit the formation or eradicate established biofilms” to “Depending on the stage of biofilm development, the AMPs may inhibit the formation of biofilm - or eradicate established biofilms”.

Supplementary materials (L645-649): Only the Figure titles should be listed (not the complete figure headings). The title of Figure S2 is missing.

Supplementary materials (page 2): According to the journal´s author guidelines, “Citations and References in Supplementary files are permitted provided that they also appear in the reference list of the main text”. The reference Riss et al. (2019) should therefore also be referred to in the manuscript.

Author Response

We thank the reviewer for the second round. All corrections suggestions were made:

  1. L440-442. Of course, the block- is more potent than the co-peptide. It is clear from the next sentence.
  2.  "in vitro" is formatted in Italic here and several more times throughout the text.
  3. The sentence "Depending on the stage of development" is corrected on L517.
  4. Supplementary materials (L630-634): the figures' titles are corrected.
  5. The reference S1 is added to the references on the main file of the manuscript.